# LC-MS/MS-Based Serum Protein Profiling for Identification of Candidate Biomarkers in Pakistani Rheumatoid Arthritis Patients

**DOI:** 10.3390/life12030464

**Published:** 2022-03-21

**Authors:** Sidrah Jahangir, Peter John, Attya Bhatti, Muhammad Muaaz Aslam, Javaid Mehmood Malik, James R. Anderson, Mandy J. Peffers

**Affiliations:** 1Atta-ur-Rahman School of Applied Biosciences (ASAB), National University of Sciences and Technology (NUST), Islamabad 44000, Pakistan; sidrah.malik@ymail.com (S.J.); attyabhatti@gmail.com (A.B.); 2Department of Human Genetics, University of Pittsburgh, Pittsburgh, PA 15216, USA; mma97@pitt.edu; 3Arthritis Research Center, Rahmat Noor Clinic, Rawalpindi 46000, Pakistan; jmmalik23@yahoo.com; 4Department of Musculoskeletal and Ageing Science, Institute of Life Course and Medical Sciences, University of Liverpool, William Henry Duncan Building, 6 West Derby Street, Liverpool L7 8TX, UK; james.anderson@liverpool.ac.uk (J.R.A.); m.j.peffers@liverpool.ac.uk (M.J.P.)

**Keywords:** rheumatoid arthritis, serum, proteomics, biomarkers, LC-MS

## Abstract

Rheumatoid arthritis is an autoimmune disorder of complex disease etiology. Currently available serological diagnostic markers lack in terms of sensitivity and specificity and thus additional biomarkers are warranted for early disease diagnosis and management. We aimed to screen and compare serum proteome profiles of rheumatoid arthritis serotypes with healthy controls in the Pakistani population for identification of potential disease biomarkers. Serum samples from rheumatoid arthritis patients and healthy controls were enriched for low abundance proteins using ProteoMiner^TM^ columns. Rheumatoid arthritis patients were assigned to one of the four serotypes based on anti-citrullinated peptide antibodies and rheumatoid factor. Serum protein profiles were analyzed via liquid chromatography-tandem mass spectrometry. The changes in the protein abundances were determined using label-free quantification software ProgenesisQI^TM^ followed by pathway analysis. Findings were validated in an independent cohort of patients and healthy controls using an enzyme-linked immunosorbent assay. A total of 213 proteins were identified. Comparative analysis of all groups (false discovery rate < 0.05, >2-fold change, and identified with ≥2 unique peptides) identified ten proteins that were differentially expressed between rheumatoid arthritis serotypes and healthy controls including pregnancy zone protein, selenoprotein P, C4b-binding protein beta chain, apolipoprotein M, N-acetylmuramoyl-L-alanine amidase, catalytic chain, oncoprotein-induced transcript 3 protein, Carboxypeptidase N subunit 2, Apolipoprotein C-I and Apolipoprotein C-III. Pathway analysis predicted inhibition of liver X receptor/retinoid X receptor activation pathway and production of nitric oxide and reactive oxygen species pathway in macrophages in all serotypes. A catalogue of potential serum biomarkers for rheumatoid arthritis were identified. These biomarkers can be further evaluated in larger cohorts from different populations for their diagnostic and prognostic potential.

## 1. Introduction

Rheumatoid arthritis (RA) is an autoimmune disorder of complex disease etiology. RA leads to the inflammation of joints and surrounding synovial membrane [1]. The global prevalence rate of RA is 0.24% and RA has been ranked as the 42nd highest contributor to global disability [2]. Diagnosing RA is a highly individualized process and is based on a combination of both clinical manifestations and serological assays. Early disease diagnosis is the key to prevent joint damage and permanent physical disability in RA [3].

RA is considered to be a continuum that begins with a disease-susceptibility stage characterized by a combination of genetic risk factors. This stage proceeds through a pre-clinical stage before the development of early RA characterized by articular inflammation. Environmental and microbial triggers continuously operate across this continuum. Immune-mediated etiology associated with stromal tissue dysregulation contributes to the chronic inflammation and ultimate articular destruction that is identified as established RA [4,5]. A number of proteins and pathways have been linked to the disease pathogenesis of RA. However, there are still some gaps in current knowledge. Research aimed at the better clarification of these mechanisms can enable the development of more specific disease-modifying therapies [6]. 

Rheumatoid factor (RF) and anti-citrullinated peptide antibodies (ACPA) are considered as the main serological markers for RA that have been included in the 2010 American College of Rheumatology (ACR)/European League against Rheumatism (EULAR) classification criteria for RA [7,8,9]. Based on 2010 ACR/EULAR classification criteria for RA, clinically diagnosed RA patients can be categorized into four serotypes: (i) positive for both RF and ACPA, (ii) positive for RF and negative for ACPA, (iii) negative for RF and positive for ACPA and (iv) negative for both RF and ACPA. However, the levels of RF are also perturbed in connective tissue diseases [10] and some chronic infectious diseases such as hepatitis B and hepatitis C virus infections [11]. RF is thus not a specific diagnostic marker for RA. ACPA is comparatively a more specific biomarker and two-thirds of the individuals ultimately diagnosed with RA were tested positive for ACPAs 6–10 years before diagnosis [12,13]. A total of 1–3% of the healthy population may also test positive for ACPAs suggesting the decreased specificity of this biomarker [14,15,16,17]. Therefore, it is important to discover the biomarkers for the diagnosis of RA with both increased sensitivity and specificity.

Mass spectrometry (MS)-based serum proteomics has emerged as a powerful technology in biological research targeted at the RA biomarker discovery [18,19]. Several proteins and peptides have been identified that are unique to serum proteome of RA patients [18,20]. A recent study compared the serum proteome profiles of seronegative patients with healthy controls [21]. However, to our knowledge, no study has compared the serum proteome profiles of all the RA serotypes based on ACPAs and RF. Furthermore, the proteomic profiles of Pakistani RA patients have not been investigated in any previous study. This study aims to screen the RA serotypes, based on ACPAs and RF, and compare them with healthy controls in the Pakistani population for the identification of biomarkers that are differentially expressed (DE) between RA patients and healthy controls.

## 2. Materials and Methods

### 2.1. Study Subjects and Serum Collection

The study was approved by the institutional review board (IRB) of the National University of Sciences and Technology (NUST), Islamabad, Pakistan, and written informed consent was obtained from all the study subjects. Human blood sera were collected from Pakistani RA patients that were diagnosed according to 2010 ACR/EULAR criteria [7] as well as healthy controls. The venous blood was collected from each patient in a 5 mL BD Vacutainer^®^ tubes (BD vacutainer TM, Frankin Lakes, NJ, USA) containing spray-coated silica and a polymer gel for serum separation. Butterfly needle was used depending on the condition of the patient. The samples were allowed to clot, and the serum was carefully alliquoted and stored at −80 °C. ACPA-status was evaluated using the commercial ACPA AESKULISA^®^ enzyme-linked immunosorbent assay (ELISA) assay kit (AESKU.Diagnostics, Wendelsheim, Germany). RF-status was determined using a latex agglutination slide test kit for RF (Werfen, Barcelona, Spain). A total of 18 patients (mean age ± SD = 40.1 ± 12.13) selected for the study were divided into 4 cohorts. The first cohort included RA patients that were double-positive for both RF and ACPA (*n* = 5), the second and third cohort included RA patients that were either positive for RF or ACPA (*n* = 5 each) and the fourth cohort included RA patients that were negative for both of these serological markers (*n* = 3). A total of 5 healthy controls *(n* = 5) (mean age ± SD = 43.4 ± 9.11) were also included in the study. Each cohort contained age-matched samples with a female-to-male ratio of 4:1. Blood samples from both RA cases and healthy controls were collected in vacutainers without anticoagulants. Serum was then separated from blood at 4000× *g* for 5 min, aliquoted into polyethylene tubes (Eppendorf AG, Hamburg, Germany) and stored at −80 °C until use.

For validation, serum samples were collected and processed from RA patients (*n* = 60) (mean age ± SD = 41.495 ± 12.8275) and healthy controls (*n* = 20) (mean age ± SD = 45.4 ± 11.31) from the same population. The demographics and clinical characteristics of the experimental and validation cohort are shown in Table 1.

### 2.2. Protein Assay

Serum samples were thawed on ice followed by centrifugation at 14,000× *g* for 10 min at 4 °C. Protein concentrations for serum samples from each donor were then determined through Pierce^®^ 660 nm protein assay kit for protein concentration (Thermo Scientific, Waltham, MA, USA). The sample volumes containing 10 mg total protein were calculated and mixed with double-distilled water (ddH_2_O) to make the total volume up to 500 µL.

### 2.3. SDS-PAGE and Silver Staining

Serum samples were analyzed using one-dimensional (1D) sodium dodecyl sulfate polyacrylamide gel electrophoresis (SDS-PAGE) for assessment of the gross quantitative as well as qualitative differences in the serum protein profiles of the study subjects. Briefly, 16 µg of serum samples were mixed with an equal volume of NativePAGE™ sample buffer (Thermo Scientific, Waltham, MA, USA) and loaded on NativePAGE™ 1.0 mm, 4–16%, bis-tris, mini protein gels (Thermo Scientific, Waltham, MA, USA). Novex Sharp Pre-Stained Protein Standard for molecular weight estimation (Thermo Scientific, Waltham, MA, USA) was also loaded in a separate well. The samples and the standard were run in NuPAGE™ MES SDS running buffer (Thermo Scientific, Waltham, MA, USA) at 120 V for 60 min and then at 150 V for 30 min. The gels were washed for 5 min in ddH_2_O. The washing was repeated thrice. Prior to visualization, the protein gels were stained for 16 hours in Coomassie Brilliant Blue R-250 dye (Bio-Rad, Hemel Hempstead, UK) and rinsed in ddH_2_O for 30 min. The whole figure can be found at Appendix A.

For qualitative assessment of the elution efficiency of ProteoMiner™ columns (Bio-Rad, Hemel Hempstead, UK), one serum sample processed through the column was also evaluated using 1D SDS-PAGE. For this purpose, the serum sample, the flow-through after each wash, and the eluted samples were run using the aforementioned protocol. Additionally, trypsin digested samples were also analyzed using 1D SDS-PAGE to confirm complete protein digestion before liquid chromatography-tandem mass spectrometry (LC–MS).

### 2.4. ProteoMiner^TM^ Column Processing

ProteoMiner™ Small Capacity bead columns for protein enrichment were loaded with 10 mg of protein from each sample separately. The bead columns were then rotated at the room temperature for 2 h followed by centrifugation at 1000× *g* for 60 s. Washing of the beads was performed thrice in phosphate-buffered saline (Sigma-Aldrich, Gillingham, UK) followed by rotation for 5 min and subsequent centrifugation for 60 s at 1000× *g*. This eluted the maximum amount of unbound protein.

### 2.5. Protein Digestion

A pre-mixed solution of 0.05% (*w*/*v*) RapiGest (Waters, Elstree, Hertfordshire, UK) and 160 µL of 25 mM ammonium bicarbonate (NH_4_HCO_3_) (Fluka Chemicals Ltd., Gillingham, UK) was used for resuspension of the Proteominer^TM^ beads. The resuspended beads were then heated for 10 min at 80 °C; DL-Dithiothreitol (Sigma-Aldrich, Gillingham, UK) to 3 mM final concentration was added, incubated for 10 min at 60 °C and iodoacetamide (Sigma-Aldrich, Gillingham, UK) was added to a final concentration of 9 mM, incubated in the dark for 30 min at room temperature. Protease enzyme trypsin (Sigma-Aldrich, Gillingham, UK) was used for enzymatic protein digestion. A total of 2 µg of trypsin was added to each sample and rotated at 37 °C for 16 h. The samples containing the beads were supplemented again with 2 µg trypsin and rotation for 2 h at 37 °C. The digested serum samples were then centrifuged at 1000× *g* for 1 min at room temperature. Supernatant was removed followed by the inhibition of the trypsin activity by acidification with 0.5% (*v*/*v*) trifluoroacetic acid (TFA, Sigma-Aldrich, Gillingham, UK) and rotation at 37 °C for 30 min. The samples were then centrifuged at 13,000× *g* for 15 min at 4 °C. 

### 2.6. LC-MS/MS

Each serum digest sample was analyzed using LC-MS/MS on an UltiMate 3000 Nano LC System (Dionex/Thermo Scientific, Waltham, MA, USA). The system was attached to a Q Exactive^TM^ Quadrupole-Orbitrap instrument (Thermo Scientific, Waltham, MA, USA). Prior to loading onto the instrument, the samples were carefully randomized using Microsoft Excel. All the samples were run in one single batch. For this purpose, 150 ng of the tryptic digest from each trypsin-digested serum sample was subjected to LC-MS/MS via a 90 min gradient. For loading on trapping column (100 Å, 75 µm × 2 cm, Acclaim PepMap 100 C18, 3 µm packing material) loading buffer was used that contained 2% (*v*/*v*) acetonitrile and 0.1% (*v*/*v*) TFA in water. The sample digests mixed with loaded buffer were run at a flow rate of 12 µL min^−1^ for 7 min. Then, a trapping column was coupled with an analytical column (100 Å, 75 µm × 50 cm, EASY-Spray PepMap RSLC C18, 2 µm packing material) followed by elution of the peptides through a linear gradient. The linear gradient consisted of 96.2% A composed of 0.1% (*v*/*v*) formic acid: 3.8% B consisting of 0.1% (*v*/*v*) formic acid in water/acetonitrile [80/20] (*v*/*v*) to 50% A: 50% B at a flow rate of 300 nl min^−1^ over 90 min and washed for 5 min at 1% A: 99% B. The column was then re-equilibrated to the starting conditions and maintained at 40 °C before direct introduction of the affluent into the integrated nano-electrospray ionization source that was operating in the positive ion mode. The MS instrument was operated in the data-dependent acquisition (DDA) mode with the survey scans between the mass to charge ratio (*m*/*z)* range of 350 to 2000 that were acquired at a mass resolution of about 60,000 and the fullwidth at half-maximum (FWHM) at *m*/*z* of about 200. The automatic gain control was set to 3e6 with a maximum injection time of 100 ms. For MS/MS, 12 of the most intense precursor ions with an isolation window of 2 *m*/*z* units and charge states ranging from 2+ to 5+ were selected. For this, the automatic gain control was set to a value of 1e5 with the maximum injection time of 100 ms. The peptide fragmentation was obtained by the higher-energy collisional dissociation utilizing a normalized collision energy of 30%. Dynamic exclusion of the *m*/*z* values was used to avoid the repeated fragmentation of the same peptide with an exclusion time of 20 s. All MS raw files for this experiment have been deposited to the ProteomeXchange Consortium through the PRIDE partner proteomics repository. The dataset identifier for this submission is PXD020235 and 10.6019/PXD020235 [22].

### 2.7. Label-Free Quantification

For label-free quantification, all the raw files were processed using Progenesis™ QI 2.0 software (Nonlinear Dynamics, Waters). Progenesis™ QI software undertakes the spectral alignment, consistent peak picking across all runs, normalization of the total protein abundance as well as peptide/protein quantification. For each feature, the top five spectra were exported, and the peptide and protein identifications were carried out via in-house Mascot server (Version 2.6.2). Reviewed *Homo sapiens* database was used to perform the identifications. Search parameters included: fragment mass tolerance value of 0.01 Da; peptide mass tolerance value of 10.0 ppm; enzyme, trypsin; one allowed missed cleavage; carbamidomethylation (cysteine) as the fixed modifications and oxidation (methionine) as the variable modification; The criteria used for protein identification included a false discovery rate (FDR) of 1% and ≥2 unique peptides.

### 2.8. Pathway Analysis

Canonical pathways, networks and disregulated regulators of the proteins that were identified with an FDR adjusted *p*-value of <0.05 and ≥2 unique peptides were performed using Ingenuity Pathway Analysis (IPA) (Qiagen, Hilden, Germany). For this, the gene names for the identified proteins were uploaded and analyzed for humans. All identified proteins were used as a background. The uncharacterized proteins were excluded from analysis.

### 2.9. Validation of MS Using ELISA

A human PZP ELISA kit (CSB-EL019131HU, CUSABIO, Houston, TX, USA) was used for the quantification of PZP protein in human samples from an independent cohort of RA patients and controls according to the manufacturer’s directions. All the samples were analyzed in duplicates and protein concentration was determined as an average of the duplicates.

### 2.10. Statistics

Heat map plots were created and visualized using MetaboAnalyst 4.0. Principal component analysis (PCA) was also performed using MetaboAnalyst 4.0. Log transformation and Pareto scaling were applied for data analysis of the normalized data. For this study, the DE proteins were defined as those with a FDR adjusted *p*-value of <0.05, identified ≥2 unique peptides and a >2 fold expression change using ANOVA. For comparison of PZP concentration between RA patients and healthy controls, a *t*-test was used. A boxplot depicting the ELISA results was designed using R 4.1.1.

## 3. Results

### 3.1. 1-D SDS-PAGE Qualitative Analysis

1-D SDS PAGE did not demonstrate any significant differences among groups (Figure 1). A large band of serum albumin appeared at 67 kDa in all the samples; the most abundant protein in human serum. 1-D SDS-PAGE of the serum samples processed through ProteoMiner™ columns showed that with each wash, the albumin and other high abundance proteins gradually decreased, and all the on-bead proteins were enriched gradually as depicted by the presence of all protein bands and their respective intensities in the SDS-PAGE of eluted samples (Figure 2).

### 3.2. Identification of Proteins in Serum

A total of 213 proteins were identified following ProgenesisQI™ using Mascot (Appendix A). One RF-negative and ACPA-positive sample returned a very low alignment score of 8.6% and was, therefore, excluded from the analysis. For the remaining samples, more than 1 unique peptide was mapped to 165 proteins out of 213 proteins. Out of 213 proteins, 124 proteins showed >a 2-fold change. A total of 37 out of these 213 proteins had q-value < 0.05.

### 3.3. Differentially Expressed Proteins

The comparative analysis of all groups (a FDR adjusted *p*-value of <0.05, identified ≥2 unique peptides and a >2 fold expression change) identified 25 proteins that were DE (Table 2), of which 10 proteins were DE between healthy control subjects and 1 of the serotypes including PZP, selenoprotein P (SELENOP), C4b-binding protein (C4BP) beta chain, apolipoprotein M (ApoM), N-acetylmuramoyl-L-alanine amidase (NAMLAA), carboxypeptidase N (CPN) catalytic chain, oncoprotein Induced Transcript 3 (OIT3), CPN subunit 2, apolipoprotein C-I (ApoC1) and apolipoprotein C-III (ApoCIII).

The PCA analysis (Figure 3A,B) showed that only 22.1% of the proteins (PC1) were divided between RA patients and healthy controls. The distribution only decreased to 21.3%, when only the patient groups were included in PCA (Figure 3C). The heat map of the proteins showed that the group averages of various proteins were different between patients and healthy subjects (Figure 4A). The heat map of the patient serotypes and controls however showed that although distinguishable patterns of expression existed between normalized abundances of individual proteins between patient serotypes as well as healthy subjects, only Q96PD5 (NAMLAA) showed similar trends across all the RA serogrpups as compared to healthy controls (Figure 4B).

### 3.4. Pathway Analysis

Canonical pathway analysis was undertaken on the DE proteins between each serotype of RA and healthy controls. The comparison of double-positive RA samples with healthy controls predicted activation of dendritic cell maturation (*p* = 0.009); and inhibition of liver X receptor/retinoid X receptor (LXR/RXR) pathway (*p* = 7.9 × 10^−^^28^), acute phase response signalling (*p* = 3.16 × 10^−^^27^) and production of NO and ROS species in themacrophages (*p* = 1.41 × 10^−^^08^) (Figure 5A). The comparison of RF-positive RA patients with healthy controls revealed an activation of the coagulation system (*p* = 3.98 × 10^−^^11^), the intrinsic prothrombin activation pathway (*p* = 8.70 × 10^−^^09^) and the GP6 signaling pathway (*p* = 0.0009); and inhibition of the LXR/RXR pathway (*p* = 5.01 × 10^−^^21^), production of NO and ROS in macrophages (*p* = 2.57 × 10^−^^08^) and maturity onset diabetes of young (MODY) signaling (*p* = 2.29 × 10^−^^06^) (Figure 5B). The comparison of ACPA-positive RA patients with healthy controls revealed activation of the coagulation system (*p* = 3.54 × 10^−^^08^), the intrinsic prothrombin activation pathway (*p* = 4.89 × 10^−^^06^), the extrinsic prothrombin activation pathway (*p* = 5.01 × 10^−^^10^) and acute phase response signalling (*p* = 5.01 × 10^−^^11^); and inhibition of the LXR/RXR pathway (*p* = 1.99 × 10^−^^14^) and production of NO and ROS in macrophages (*p* = 0.001) (Figure 5C). Pathway analysis of double-negative RA patients with healthy controls revealed the activation of the coagulation system (*p* = 7.94 × 10^−^^19^), the intrinsic prothrombin activation pathway (*p* = 5.01 × 10^−^^12^) and the extrinsic prothrombin activation pathway (*p* = 1.25 × 10^−^^13^); and inhibition of the LXR/RXR pathway (*p* = 1.58 × 10^−^^25^); acute phase response signalling (*p* = 1 × 10^−^^23^) and production of NO and ROS in macrophages (*p* = 2.18 × 10^−^^10^) (Figure 5D).

The comparison of the four serotypes of RA with healthy controls revealed an inhibition of inflammatory response, leukocyte migration, binding of professional phagocytic cells, migration of cells, adhesion of phagocytes, cell movement of phagocytes and cell movement of leukocytes in all serotypes except double-negative serotype. Accumulation of leukocytes was, however, inhibited in all serotypes. Concentration of cholesterol was inhibited in all serotypes except ACPA-positive patients that did not show activation or inhibition of this protein (Figure 6). The detailed results of pathway analysis are provided in Appendix A.

### 3.5. Validation of Mass Spectrometry Using ELISA

We validated the mass spectrometry results using ELISA for PZP. As Figure 7 shows, the expression of PZP was significantly higher among patients (7.54 ± 6.35 µg/mL) as compared to controls (1.03 ± 0.54 µg/mL (*p*-value 7.41 × 10^−11^). The PZP concentration for each sample is represented in Appendix A. The sensitivity of PZP for detecting RA is 96.7% and specificity is 95%.

## 4. Discussion

In this study, we identified 10 DE proteins between RA serotypes and healthy controls. Next, we undertook successfully validation of one of the DE proteins; PZP, in an independent sample cohort indicating our findings for this protein are applicable to another population. We then performed canonical pathway analysis for the DE proteins across each serotype in comparison to healthy controls to identify the key pathways and biological processes that are perturbed across these serotypes. We used ProteoMiner^TM^ protein enrichment columns to deplete the proteins with high abundance and enrich the proteins with low abundance [23]. ProteoMiner^TM^ protein enrichment of low abundance proteins has several advantages over the immunoaffinity-based protein depletion approaches including a relatively less-complicated procedure, high material yield and reproducibility [24,25].

PZP is a high-molecular-weight immunosuppressive glycoprotein that is elevated during pregnancy. The role of this protein as an autoimmunity mediator was established by a recent LC–MS/MS-based study in inflammatory bowel disease patients [26]. In this study, we also found increased expression of PZP in all the RA serotypes as compared to the controls using LC–MS/MS. The results were further validated by ELISA in a different cohort of RA patients and subjects. The high sensitivity and specificity of this protein for RA patients signify strong candidacy of PZP as a disease biomarker.

In this study, the serum expression of SELENOP was decreased in all RA serotypes in comparison to controls. SELENOP is a biomarker of selenium status that has been identified as a major preventable trigger for autoimmune diseases including RA [27]. In comparison to controls, the serum selenium concentrations [28] and SELENOP concentrations [29,30] have been reported to be decreased in RA patients. The selenium status has been linked to the upregulation of a whole set of inflammation-related genes via nuclear factor kappa-light-chain enhancer of activated B cells (NF-κB) mediated activation of several intracellular selenoproteins [28]. The role of selenium and SELENOP, combined with previous findings suggest strong candidacy of this protein as a biomarker of autoimmunity.

NAMLAA degrades bacterial cell wall component peptidoglycan [31] that has strong pro-inflammatory properties and can induce arthritis in rat models [32,33]. The degradation of these pro-inflammatory components should suggestively confer an anti-inflammatory and protective role to NAMLAA against arthritis. However, Saha et al. [34] demonstrated that NAMLAA is indeed essential for the development of arthritis, a relatively unexpected finding. The study findings of Saha et al. [34] have not been supported by animal model studies for other inflammatory diseases [35]. Decreased levels of this protein in human RA subjects as compared to healthy controls were observed in this study. The autoantigenic potential of NAMLAA and the presence of antibodies has been reported in a recent study [18] that can explain the lower serum levels of circulating NAMLAA. The imbalance of this homeostasis is probably responsible for the development of RA that needs to be further explored. 

C4BP β-chain, a complement inhibitor [36], and CPN, a zinc metalloprotease [37], were also observed to be DE in this study. However, a lack of consensus regarding the role of these proteins in autoimmunity and RA hereby suggest further exploration.

We found three apolipoproteins to be DE between RA patients and healthy controls including ApoM, ApoC1 and ApoCIII. These apolipoproteins are implicated in protection against atherosclerosis owing to their role in HDL metabolism as well as anti-inflammatory properties [38]. The polymorphisms in the ApoM gene have been associated with the risk of dyslipidaemia in RA patients [39,40]. However, no study reports the serum levels of this chaperone in RA patients. ApoC1 has been identified as a predictor of drug response to RA [41,42]. The risk of developing cardiovascular disease is elevated among RA patients than the general population [43,44]. The observed decrease in the serum levels of these apolipoproteins in RA patients could suggestively explain the increased risk of developing cardiovascular disease among RA patients and highlight the link between these two illnesses.

The pathway analysis of the DE proteins showed that some pathways were differentially inhibited or activated in various serotypes suggesting that these serotypes are indeed regulated by different pathogenic mechanisms. However, some similarities were also observed including inhibition of LXR/RXR pathway and NO and ROS production in macrophages. LXR/RXR pathway was inhibited among all the RA serotypes. This pathway has been reported to inhibit atherosclerosis [45] and inflammation [46], suggesting an important and relatively unexplored link between this pathway and RA. The role of ROS in autoimmunity is complex and has been generally viewed as detrimental in the pathogenesis of autoimmune disease [47]. A recent study revealed the regulatory role of these oxidative stress markers to prevent the pathogenesis of chronic inflammatory diseases [48]. The inhibition of NO and ROS pathway in macrophage across all the serotypes warrants further exploration about the precise role of this pathway in the pathogenesis of RA.

RA is a complex disorder with molecular and clinical heterogeneity. We used RF and ACPA to classify our patient population and studied the DE proteins in comparison to all healthy controls. However, due to the COVID-19 pandemic, only a limited number of samples could be collected for validation of the identified proteins. The lockdown situation also limited the access to the laboratory facilities and the samples were not tested for their individual RF and ACPA status. The validation of the mass spectrometry result for PZP in an independent cohort of patients suggest that identified proteins can be tested on larger cohorts of patients from different populations in the future to validate the study findings and identify disease biomarkers for RA.

## 5. Conclusions

RA is a complex disease that is influenced by an intricate interactome of various environmental, genetic and microbial factors that influence the immune homeostasis. Owing to the complex genetic architecture accompanied by a plethora of microbial and environmental triggers that an organism is exposed to this has made the identification of diagnostic and prognostic markers challenging. Our study has explored the serum proteomics of this complex autoimmune disorder in a relatively understudied Pakistani population to identify disease biomarkers that are DE among various serotypes of RA patients and healthy controls. We identified that PZP, SELENOP, C4BP beta chain, ApoM, NAMLAA, CPN catalytic chain, OIT3, CPN subunit 2, ApoC1 and ApoCIII were DE between the RA patients and healthy controls. These serum proteins have strong potential to serve as diagnostic and prognostic biomarkers of RA and can also be evaluated to fill the gaps in the current knowledge of pathogenesis of RA. These findings can be validated in larger cohorts from different populations to identify diagnostic and prognostic biomarkers of RA.

## Figures and Tables

**Figure 1 life-12-00464-f001:**
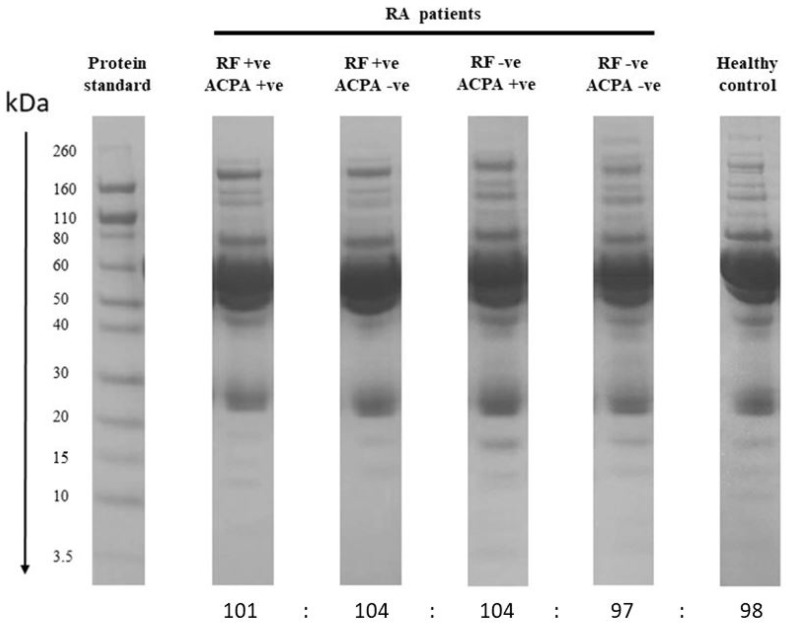
Coomassie-stained 1D SDS-PAGE of serum samples from RF-positive and ACPA-positive, RF-positive and ACPA-negative, RF-negative and ACPA-postive and RF-negative and ACPA-negative RA patients along with healthy control. Equal weight of protein (16 μg per well) from all serum samples was used for qualitative comparison of the serum proteome profiles of the samples from different serotypes and healthy control. The vertical black line indicates molecular weight.

**Figure 2 life-12-00464-f002:**
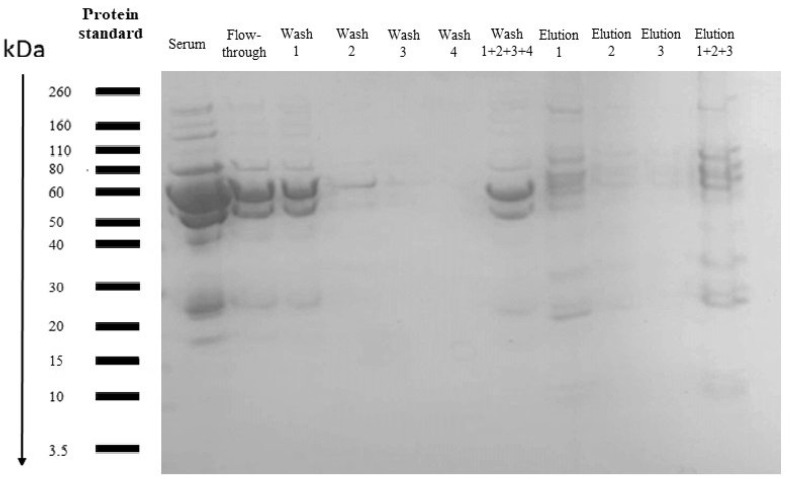
Coomassie-stained 1D SDS-PAGE of serum sample loaded on ProteoMiner™ SmallCapacity bead columns; flow-through after each wash, flow-through combined and bead-bound protein after three elutions indicate that all the proteins were gradually enriched.

**Figure 3 life-12-00464-f003:**
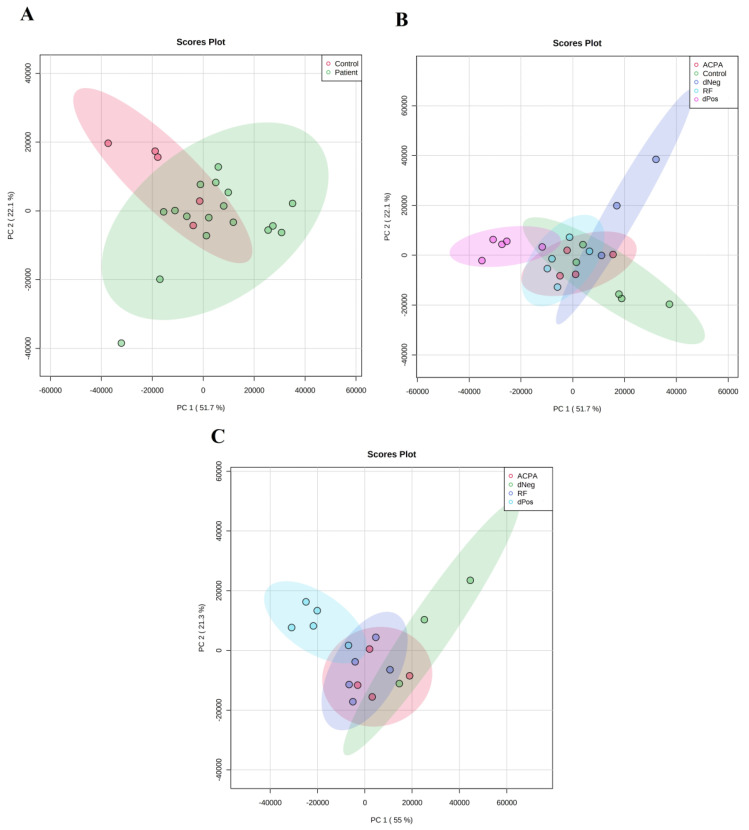
Principal component analysis plots of identified proteins (a FDR adjusted *p*-value of <0.05, identified ≥2 unique peptides and a >2 fold expression change). (**A**) Healthy versus RA; (**B**) healthy controls and various serotypes of RA, including both RF and ACPA positive (RF ACPA), only RF positive (RF), only ACPA positive (ACPA) and negative for both RF and ACPA (dNeg) (**C**) RA serotypes only.

**Figure 4 life-12-00464-f004:**
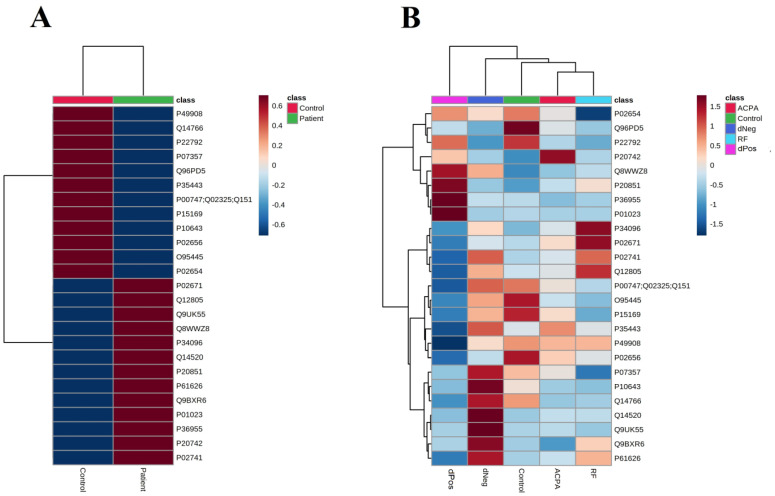
Heat map of 25 differentially expressed proteins between RA patients and healthy controls (an FDR-adjusted *p*-value of <0.05, identified ≥ 2 unique peptides and a > 2-fold expression change (**A**) The expressed group averages of the healthy versus RA (**B**) The expressed group averages of the healthy controls and various serotypes of RA, including both RF and ACPA positive (RF ACPA), only RF positive (RF), only ACPA positive (ACPA) and negative for both RF and ACPA (dNeg). Each row represents one protein. The red shading represents an increase in the expression and the blue shading represents a decreased expression. Images made in Metaboanalyst.

**Figure 5 life-12-00464-f005:**
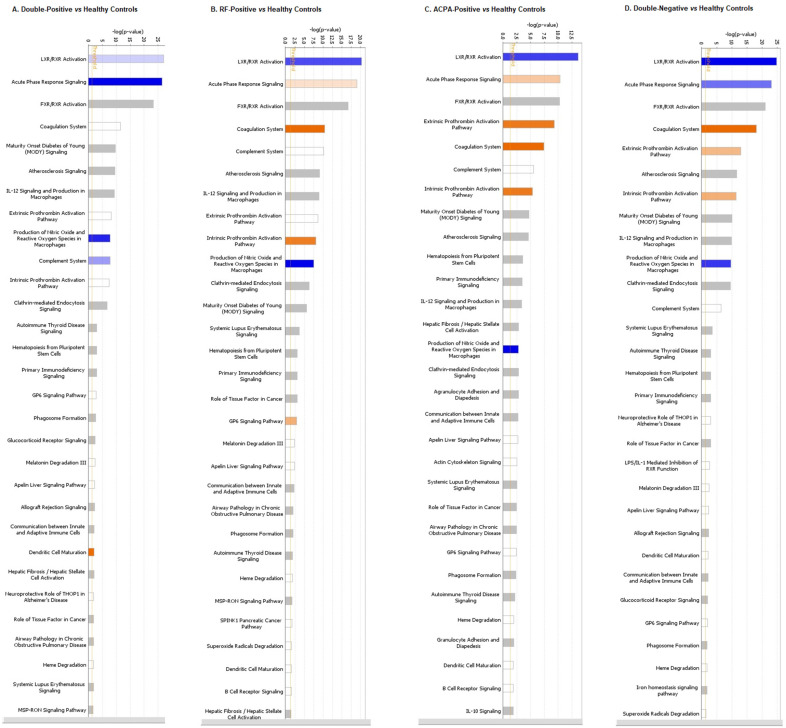
Canonical pathways identified through IPA analysis of the differentially abundant proteins in the serum samples of (**A**) double-positive RA patients (**B**) RF-positive RA patients (**C**) ACPA-positive RA patients (**D**) double-negative RA patients compared to the healthy subjects. Each bar hereby represents the significance of the individual canonical pathway associated with the serotype. The significance has been calculated by using a right-sided Fisher’s exact test. Orange bars hereby represent upregulated canonical pathways and blue bars represent the down-regulated pathways.

**Figure 6 life-12-00464-f006:**
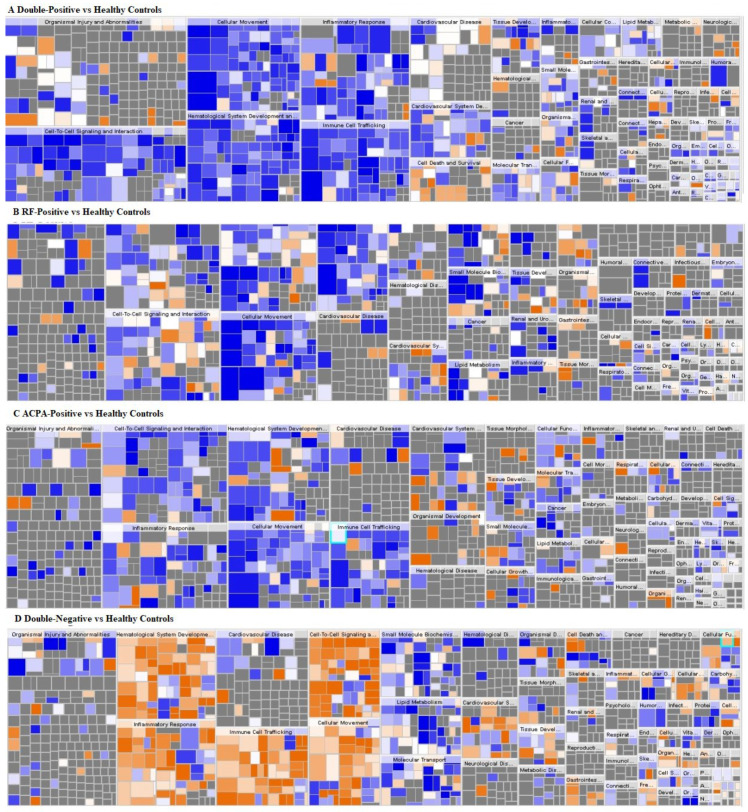
Heat map identifications of the canonical pathway groupings for identification of the altered molecular and the cellular functions in: RA patients (**A**), RF negative and ACPA negative (**B**), ACPA positive (**C**), RF positive (**D**), RF negative and ACPA negative compared to healthy subjects. The colors of the squares represent the Z score. The positive Z score represents a positive association of the proteins up- or downregulation with the activation of the respective function. The negative Z score on contrary represents inhibition of the function. The orange-colored squares represent upregulation during the disease state and the blue squares represent downregulation with the color intensity being directly correlated with the prediction strength.

**Figure 7 life-12-00464-f007:**
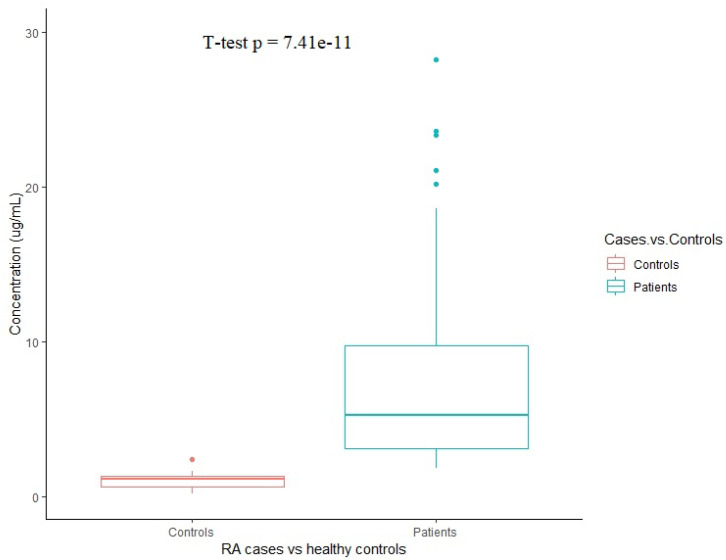
The serum levels of human PZP proteins quantified using ELISA in RA patients (*n* = 60) and healthy controls (*n* = 20). *t*-test was used to compare the serum concentration of PZP. Boxplot was designed using R 4.1.1.

**Table 1 life-12-00464-t001:** Demographics and clinical characteristics of the rheumatoid arthritis patients and healthy controls.

	Double Positive	RF Positive	ACPA Positive	Double Negative	Healthy
**Discovery cohort**					
Number of subjects (*n*)	5	5	5	3	5
Sex (Male/Females)	4/1	4/1	4/1	2/1	4/1
Age	40.77 ± 12.01	39.31 ±13.43	41.21 ± 12.02	39.13 ± 11.07	43.4 ± 9.11
DAS 28	4.93 ± 1.13	4.41 ± 0.91	3.98 ± 1.11	2.31 ± 1.73	-
Treatment					
Methotrexate (*n*)	1	1	2	2	-
Hydroxychloroquine + methotrexate + sulfasalazine (*n*)	3	2	2	1	-
Sulfasalazine + hydroxychloroquine (*n*)	-	1	-	-	-
Leflunomide + methotrexate (*n*)	1	1	1	-	-
Comorbidities					
Hypertension (*n*)	2	1	-	-	1
Type 1 diabetes (*n*)		1	2	1	-
**Validation cohort**					
Number of subjects (*n*)	20	17	13	10	20
Sex(Male/Females)	16/4	13/4	11/2	8/2	16/4
Age	46.66 ± 13.44	41.03 ± 14.14	38.21 ± 11.61	40.08 ± 12.12	45.4 ± 11.31
DAS 28	3.98 ± 1.1	4.31 ± 1.32	4.11 ± 2.1	3.11 ± 2.17	-
Treatment					
Methotrexate (*n*)	5	8	3	4	-
Hydroxychloroquine + methotrexate + sulfasalazine (*n*)	7	3	3	2	-
Sulfasalazine + hydroxychloroquine (*n*)	4	3	4	3	-
Leflunomide + methotrexate (*n*)	4	2	3	1	-
Comorbidities					
Hypertension (*n*)	3	4	2	3	3
Type 1 diabetes (*n*)	4	2	2	1	3

**Table 2 life-12-00464-t002:** Differentially expressed proteins with q-value of <0.05, identified with at least two unique peptides > two-fold change between all groups.

Accession	q Value	Number of Unique Peptides	Maximum Fold Change	Highest Mean Condition	Lowest Mean Condition	Protein Name
P49908	0.0007	2	5.19	Controls	dPos	Selenoprotein P
P00747;Q02325;Q15195	0.0009	49	2.13	dNeg	dPos	Plasminogen
P20742	0.009	14	6.37	ACPA	Controls	Pregnancy zone protein
P34096	0.009	2	5.33	RF	dPos	Ribonuclease 4
P02741	0.01	4	8.91	dNeg	dPos	C-reactive protein
P20851	0.01	6	2.30	dPos	Controls	C4b-binding protein beta chain
O95445	0.02	7	3.32	Controls	dPos	Apolipoprotein M
P36955	0.02	9	2.21	dPos	ACPA	Pigment epithelium-derived factor
Q14520	0.02	15	2.06	dNeg	dPos	Hyaluronan-binding protein 2
Q96PD5	0.02	6	2.66	Controls	dNeg	N-acetylmuramoyl-L-alanine amidase
P15169	0.02	4	3.23	Controls	dPos	Carboxypeptidase N catalytic chain
P01023	0.02	10	13.4	dPos	dNeg	Alpha-2-macroglobulin
Q8WWZ8	0.02	2	3.64	dPos	Controls	Oncoprotein-induced transcript 3 protein
P35443	0.02	3	3.08	dNeg	dPos	Thrombospondin-4
P02671	0.02	7	2.86	RF	dPos	Fibrinogen alpha chain
P07357	0.02	21	2.39	dNeg	RF	Complement component C8 alpha chain
P22792	0.02	7	2.26	Control	dNeg	Carboxypeptidase N subunit 2
P02654	0.03	2	4.27	Controls	RF	Apolipoprotein C-I
Q9BXR6	0.03	17	2.43	dNeg	ACPA	Complement factor H-related protein 5
Q9UK55	0.031311	8	5.59	dNeg	RF	Protein Z-dependent protease inhibitor
P61626	0.03	5	3.70	dNeg	dPos	Lysozyme C
P02656	0.03	5	2.48	Controls	dPos	Apolipoprotein C-III
P10643	0.03	18	2.47	dNeg	dPos	Complement component C7
Q14766	0.04	4	5.30	dNeg	dPos	Latent-transforming growth factor beta-binding protein 1
Q12805	0.04	11	2.46	RF	dPos	EGF-containing fibulin-like extracellular matrix protein 1

## Data Availability

The MS raw data for this study are available at the ProteomeXchange Consortium doi PXD020235, 10.6019/PXD020235.

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
