# Peer review of "LC-MS/MS-Based Serum Protein Profiling for Identification of Candidate Biomarkers in Pakistani Rheumatoid Arthritis Patients"

_life, 2022, doi:10.3390/life12030464_

Round 1

Reviewer 1 Report

The authors collected serum samples from different subtypes of RA patients in this manuscript. They employ LC-MS to identify and quantify the peptides in the hope of finding a peptide that is specifically associated with RA or a particular subtype of RA. Based on their analysis, PZP will be the most relevant peptide associated with RA. There are some concerns in this manuscript:

  1. Figure 1, is it 1 kDa instead of 10 kDa?
  2. The authors should mention the more detailed clinical characteristics of RF, ACPA, dPos, d Neg. Which of them will have a higher risk ratio? Also, when did the authors collect the serum samples? Before the treatment or after the treatment?
  3. Based on figure 4B, there are many up-regulated peptides in the dNeg group. However, there is no specific subgroup of peptides in the control group that is up-regulated or down-regulated. How can the authors generate figure 4B which shows a clear differential expression pattern?
  4. What is the importance of the results illustrated in figures 5-6? The results in these figures are not the supporting evidence for figure 7 and other results. Also, the manuscript is not focusing on the pathogenesis of RA.
  5. What are the sensitivity and specificity of PZP protein for detecting RA?  

Author Response

  1. Figure 1, is it 1 kDa instead of 10 kDa?

Response:

We thank the reviewer for pointing out this mistake in labelling the molecular ladder. We have amended the figures by relooking into the reference. The label of 5 was wrong. We have corrected this to 15 kDa in both Figure 1 and Figure 2.

  1. The authors should mention the more detailed clinical characteristics of RF, ACPA, dPos, d Neg. Which of them will have a higher risk ratio? Also, when did the authors collect the serum samples? Before the treatment or after the treatment?

Response:

We thank the reviewer for this comment. We have included the detailed characteristics of each serotype for both experimental and validation cohort in our manuscript.

All of the patients are clinically diagnosed patients of rheumatoid arthritis according to 2010 ACR/EULAR criteria (1).

We have collected the serum samples of the patients that visited the collaborating ‘Rheumatoid Arthritis Clinic’. The patients were diagnosed by the clinician. We have not stratified the patients according to their treatment characteristics. However, we collected this information in our data. This information has also been included in Table 1.

  1. Based on figure 4B, there are many up-regulated peptides in the dNeg group. However, there is no specific subgroup of peptides in the control group that is up-regulated or down-regulated. How can the authors generate figure 4B which shows a clear differential expression pattern?

Response:

Figure 4B shows the comparative analysis of the identified proteins from LC-MS analysis. We agree with reviewer, that it is difficult to identify a single group of proteins that shows differential expression. However, the metaboanalyst used corrected p-value for analysis and the differences identified are not as clear as found out through analysis that consider p-value, FDR and number of unique peptides (Table 2). Some proteins have shown clear differential expression, as for instance P20742. This protein has the highest levels in ACPA positive patients followed by double positive and then ACPA positive and double negative. The expression is lowest in the control group (deepest blue). If compared collectively, the serum expression is higher across patients than controls. This ID is for PZP, one of the candidate biomarkers we used for validating our findings and we found a clear difference in the levels of patients.

Similarly, Q96PD5 that encodes for NAMLAA shows highest expression in the controls and the expression is much lower across all serotypes. However, it was beyond the scope of our study to validate all these markers individually. For further clarification we have enlisted all the DE proteins in Table 2

  1. What is the importance of the results illustrated in figures 5-6? The results in these figures are not the supporting evidence for figure 7 and other results. Also, the manuscript is not focusing on the pathogenesis of RA.

Response:

We thank the reviewer for this comment and are sorry we did not explain our approach clear enough. Rheumatoid arthritis is a complex disease and various different proteins, through a complex set of interactome contribute to the pathogenesis of the disease. We compared the serum proteome profiles of rheumatoid arthritis serotypes (based on RF and ACPA status) and identified a catalogue of proteins that was differentially expressed. One of these was used for validation of the above stated results in a separate, relatively larger cohort of patients using ELISA. Figures 5 and 6 use the set of differentially expressed proteins for ‘each’ serotype in comparison to controls. For this, we used a proprietary bioinformatics pathway analysis tool the potential consequences of the differentially expressed proteins in our dataset. The robustness in this methodology is the well curated and updated tool (IPA) and the way we are able to input directionality of the protein into pathway analysis. To increase robustness, we also apply the approach of inputting only the identified proteins as background to prevent bias. In Figure 5 we show the significant canonical pathways for each group. Pathways with red or blue bars represent significant activation z scores meaning that the majority of proteins in this pathway follow a single direction indicating a predicted increase or decrease in that pathway. Figure 6 has highlighted biological processes in sets of heatmaps for the differentially expressed proteins in the study, again with z scores highlighted. In discussion we have discussed the pathways that showed similar trends across all serotypes including inhibition of LXR/RXR pathway and NO and ROS production in macrophages. We of course realise that these pathways and processes need validating in our future work but this is beyond the scope of this study. Figure 7 shows validation of one of the proteins that we found changing in our study. This is therefore looking at a single protein in the set differentially expressed whereas Figure 5-6 show potential global changes in our data.

So in summary, we identified the differentially expressed protein across all serotypes in comparison to controls (Table 2) and then we performed individual pathway analysis for differentially expressed proteins for each individual serotype with controls (Supplementary file 2 and 3). Figure 5 and 6 shows these pathways and biological processes. Figure 7 includes validation results for one of these differentially expressed proteins ‘PZP’ that were checked using ELISA. Hereby the patients show a significant increase in expression as compared to controls. This is also evident from the heat map of this protein (P20742) in figure 4. Validation in an independent cohort of subjects signifies the strength of results of LC-MS data. We have tried to clarify this approach in context of pathogenesis of rheumatoid arthritis in both introduction and discussion section.

  1. What are the sensitivity and specificity of PZP protein for detecting RA?  

Response:

We are really thankful to the reviewer for raising this point. We have now included the sensitivity and specificity of this biomarker in results section as well and discussed it in the discussion section.

Reviewer 2 Report

I consider the work interesting, well structured and balanced in its different sections. The results are presented in a set of adequate and properly descriptive and detailed figures and tables. I consider the results duly discussed and the authors assume some limitations of the work, largely due to Covid19. In this sense, perhaps the title should reflect that the results are preliminary, justifying confirmation in a larger sample.

Comment 1: My main criticism is for the sample size, and the authors assume this in the Discussion (lines 432-435). This aspect is even more emphasized if both male and female are considered for each group (which would have been preferable to do only one of the sexes).

Comment 2: Considering that RA “…is a complex disease that is influenced by intricate interactome of various environmental, genetic and microbial factors that influence the immune homeostasis…, the characterization of patients and the control group should be more detailed, if the individuals had, or no, any comorbidity and whether they were taking any medication, namely anti-inflammatory drugs.

Comment 3: How was the blood collected? And always at the same time of day?

Comment 4: What are the characteristics of validation patients in relation to reference biomarkers? RA serotypes?

Comment 5, line 38: “…RA serotypes…” »» “…Rheumatoid arthritis (RA)…”

Comment 6, line 51:  What do these percentages refer to? Global prevalence over what? “…global prevalence rate of RA is 0.5 to 1% …”

Comment 7: “…1-D SDS PAGE did not demonstrate any significant differences (Figure 1)… “ »»» “…1-D SDS PAGE did not demonstrate any significant differences among groups (Figure 1)…”

Author Response

Comment 1: My main criticism is for the sample size, and the authors assume this in the Discussion (lines 432-435). This aspect is even more emphasized if both male and female are considered for each group (which would have been preferable to do only one of the sexes).

Response:

We thank the reviewer for their insight and agree with his concern. We have changed the title to include ‘candidate’ as these markers need to be further evaluated.

We have also increased the sample size of the validation cohort. The experimental cohort increase is not possible due some technical reasons. However, validation was carried out in Pakistan, but previously when we were trying to validate, the access to samples was restricted due to COVID-19 lockdown. But now most of the restrictions have been lifted and the patients’ blood is being collected routinely for on-going projects in the lab. We have therefore included more patients in the study for validation and have performed Pregnancy Zone Protein ELISA that we initially wanted to. Now, the size of the validation cohort has increased a little.  We have tried to balance as many demographics and characteristics as possible during our study between the serotypes.

As for sex, rheumatoid arthritis is more common among females as compared to males with a female to male ratio of 3:1 (1). We have tried to maintain this high female to male ratio across all serotypes and controls in both experimental and validation cohort.

Comment 2: Considering that RA “…is a complex disease that is influenced by intricate interactome of various environmental, genetic and microbial factors that influence the immune homeostasis…, the characterization of patients and the control group should be more detailed, if the individuals had, or no, any comorbidity and whether they were taking any medication, namely anti-inflammatory drugs.

Response:

We are really thankful to the reviewer for pointing this out. We have now included a table (Table 1) to include more detailed characteristics of the various serotypes of rheumatoid arthritis patients and controls for both experimental and validation cohort.

Comment 3: How was the blood collected? And always at the same time of day?

Response:

The blood was collected from all the patients from 11:00 a.m. to 1:00 p.m. each day including both patients and controls. The blood was collected using a syringe and stored in BD vacutainer tube with spray-coated silica and a polymer gel for serum separation. For the fragile patients (especially some elderly), we used a butterfly needle was used. We have now included this information in the methodology section.

Comment 4: What are the characteristics of validation patients in relation to reference biomarkers? RA serotypes?

Response:

We are really thankful for raising this point. We have now included the detailed clinical characteristics of the patients included in the validation cohort in relation to their seropositivity status in Table 1.

Comment 5, line 38: “…RA serotypes…” »» “…Rheumatoid arthritis (RA)…”

Response:

We thank the reviewer for pointing this typo out and have removed the abbreviation from the abstract.

Comment 6, line 51:  What do these percentages refer to? Global prevalence over what? “…global prevalence rate of RA is 0.5 to 1% …”

Response:

We are really thankful for your comment. Prevalence is the proportion of a population who have a specific characteristic in a given time period. We have now updated the statistics and included a more recent and global study that state the prevalence of rheumatoid arthritis as 0.24%.   The global prevalence of RA (from 5 to 100 years of age) in 2010 was estimated to be 0.24% (95% CI 0.23% to 0.25%). The research used meta-regression tool, DisMod-MR for calculating the global burden of the disease.  RA prevalence, incidence and mortality data were entered into DisMod-MR, which pooled the available heterogeneous data to adjust for methodological differences and checked these data for internal consistency. DisMod-MR used these data to predict values for countries and regions with little or no data using disease-relevant country characteristics and random effects for country, region and super-region. The tool produced a full set of age/sex/region/ year-specific estimates for prevalence (2).

Comment 7: “…1-D SDS PAGE did not demonstrate any significant differences (Figure 1)… “ »»» “…1-D SDS PAGE did not demonstrate any significant differences among groups (Figure 1)…”

Response:

We thank the reviewer for pointing this typo out and have altered the wording in the manuscript.

  1. Aletaha D, Neogi T, Silman AJ, Funovits J, Felson DT, Bingham CO, 3rd, et al. 2010 Rheumatoid arthritis classification criteria: an American College of Rheumatology/European League Against Rheumatism collaborative initiative. Arthritis Rheum. 2010;62(9):2569-81.
  2. Cross, M., Smith, E., Hoy, D., Carmona, L., Wolfe, F., Vos, T., ... & March, L. (2014). The global burden of rheumatoid arthritis: estimates from the global burden of disease 2010 study. Annals of the rheumatic diseases, 73(7), 1316-1322.

Round 2

Reviewer 1 Report

The authors have already addressed all my concerns. The manuscript is fine for publication.